# A host beetle pheromone regulates development and behavior in the nematode *Pristionchus pacificus*

Jessica K Cinkornpumin[1†], Dona R Wisidagama[1†‡], Veronika Rapoport[1],
James L Go[1], Christoph Dieterich[2], Xiaoyue Wang[3], Ralf J Sommer[3], Ray L Hong[1*]

[1]Department of Biology, California State University, Northridge, Northridge, United States; [2]Department of Bioinformatics, Max Planck Institute for Biology of Ageing, Cologne, Germany; [3]Department for Evolutionary Biology, Max Planck Institute for Developmental Biology, Tuebingen, Germany

**Abstract** Nematodes and insects are the two most speciose animal phyla and nematode–insect associations encompass widespread biological interactions. To dissect the chemical signals and the genes mediating this association, we investigated the effect of an oriental beetle sex pheromone on the development and behavior of the nematode *Pristionchus pacificus*. We found that while the beetle pheromone is attractive to *P. pacificus* adults, the pheromone arrests embryo development, paralyzes J2 larva, and inhibits exit of dauer larvae. To uncover the mechanism that regulates insect pheromone sensitivity, a newly identified mutant, *Ppa-obi-1*, is used to reveal the molecular links between altered attraction towards the beetle pheromone, as well as hypersensitivity to its paralyzing effects. *Ppa-obi-1* encodes lipid-binding domains and reaches its highest expression in various cell types, including the amphid neuron sheath and excretory cells. Our data suggest that the beetle host pheromone may be a species-specific volatile synomone that co-evolved with necromeny.

**\*For correspondence:** ray.
hong@csun.edu

[†]These authors contributed
equally to this work

**Present address:** [‡]Department
of Human Genetics, University of
Utah, Salt Lake City, United
States

**Competing interests:** The
authors declare that no
competing interests exist.

**Reviewing editor**: Oliver Hobert,
Columbia University, United
States

## Introduction

An understanding of ecological interactions and evolutionary history can be beneficial to the understanding of the cellular and developmental processes of an organism. For example, the genes responsible for fruit and bacterial odor preferences in *Drosophila melanogaster* and *Caenorhabditis elegans* are most informative when considering the natural ecologies in rotting fruits and vegetation (*Bargmann et al., 1993*; *Bargmann, 2006*; *Hallem and Carlson, 2006*; *Kiontke and Sudhaus, 2006*). In *Pristionchus pacificus*, an emerging model organism for the study of development and behavior (*Hong and Sommer, 2006b*), a systematic effort to identify the natural ecology of *Pristionchus* nematodes revealed that *P. pacificus* host preferences include the oriental beetle (*Exomala orientalis*) found in Japan and northeastern United States (*Herrmann et al., 2007*). Unlike several fructivorous *Caenorhabditis* species, *Pristionchus* species are considered to be necromenic nematodes that have species-specific beetle host preferences and feed on the microorganisms that emerge from the beetle carcass (*Herrmann et al., 2006a, 2006b*; *Brown et al., 2011*). *Pristionchus* species are members of the Diplogasteridae family of nematodes whose common ancestor with the Rhabditidae family (e.g., *C. elegans*) diverged ~300 million years ago (*Dieterich et al., 2008*). Given that beetles (Coleoptera) represent a group with the most described species, the possibility that most beetle species can harbor specific entomophilic nematode associations underscores the vast number of nematode species remaining to be described. Although biologists have been aware of such species-specific insect–nematode associations for some time, the genetic basis for such interactions remains largely unknown.

**eLife digest** The nematode worm *Pristionchus pacificus* can live as a parasite inside the oriental beetle, where it waits for the beetle to die so it can feed off the bacteria that live on the beetle's decomposing carcass. This ecologically important interaction is called necromeny. *P. pacificus* is attracted to a new host by a sex pheromone produced by the beetle, but the genes and biological mechanisms that enable this interaction to occur are not understood in much detail.

To identify the genetic basis of this interaction, Cinkornpumin et al. identified and examined a mutant form of *P. pacificus* that cannot sense the beetle sex pheromone. This revealed that although this pheromone attracts the adult nematodes, it stops *P. pacificus* embryos developing and can paralyze larvae. Cinkornpumin et al. suggest that the pheromone has likely evolved this ability in order to counteract the spread of the nematodes. This result implies that being invaded by *P. pacificus* makes life more difficult for the beetles than was previously thought.

Further investigation of the gene damaged in the *P. pacificus* mutants revealed that it encodes a protein that may bind to molecules called lipids, which are needed to form cell membranes and are used in cell signaling. As well as helping the nematodes to detect the sex pheromone, the lipid-binding protein also appears to help protect the worms from the pheromone's detrimental effects.

Cinkornpumin et al. observed that the gene for the lipid-binding protein is activated in several tissues, including the cells that form a sheath around some of the nerves that detect chemical signals. Whether this tissue is responsible for the chemical-sensing abilities of the lipid-binding protein, and whether these same tissues are responsible for protecting the nematodes from the damaging effects of the pheromone, remains to be discovered.

Previous studies suggest that the beetle host specificity of *Pristionchus* species is in part due to nematode attraction to various plant volatile compounds released during beetle feeding, as well as to beetle aggregation and sex pheromones during mating (*Hong and Sommer, 2006a*; *Hong et al., 2008b*, *2008a*). For example, *Pristionchus maupasi* associates primarily with the European May beetle *Melolontha* spp. and is attracted to the blend of the green leaf alcohol from plants and the phenol pheromone from beetles (*Hong et al., 2008b*). Thus, insect and plant odors can synergize to provide nematodes with information on host presence, sex, and maturity. By contrast, *C. elegans* attractants such as 2,3-butanedione (diacetyl), 2,3-pentanedione, and isoamyl alcohol are significantly less attractive to *P. pacificus* (*Bargmann et al., 1993*; *Hong and Sommer, 2006a*). This piggybacking of intra-species communication for inter-species interactions between *Pristionchus* and beetles can also evolve rapidly at the population level, as exhibited by the natural variation of *P. pacificus* odor preference for insect pheromones (*Hong et al., 2008a*; *McGaughran et al., 2013b*). In particular, *P. pacificus* Washington strain (PS1843) is attracted to the lepidopteran sex pheromone *E*-11-tetradecenyl acetate (ETDA) and the oriental beetle sex pheromone *z*-7-tetradece-2-one (ZTDO) (*Herrmann et al., 2007*), but the wild-type reference strain from California (PS312) requires exogenous cGMP treatment to become attracted to these insect pheromones (*Hong et al., 2008a*). These natural variations in insect pheromone attraction, combined with regional-specific *P. pacificus* host preferences, may offer support for the geographic mosaic theory of co-evolution (*Thompson, 2005*).

Although the distinct chemosensory profiles of *P. pacificus* populations suggest insect pheromone attraction to be an important and rapidly evolving trait, other studies on insect–nematode interactions show that there are other potential roles host chemicals can play in coordinating host–parasite development and nematode dispersal behavior. For example, the pinewood nematode *Bursaphelenchus xylophilus* is intimately associated with its *Monochamus* beetle vector, and fatty acid ethyl esters exuded during beetle eclosion promote the formation of the dispersal fourth larval stage *B. xylophilus* (*Zhao et al., 2013*). Similarly, the species-specific phoretic association between the burrower bug *Parastrachia japonensis* and the nematode *Caenorhabditis japonica* may be mediated by the beetle cuticular compounds that not only attract *C. japonica* but also assist infestation by inhibiting *C. japonica* dispersal (*Okumura et al., 2013*). *P. pacificus* populations have been found on various beetle hosts worldwide, but the association with the oriental beetle was the first to be identified and the only one whose sex pheromone is a strong attractant (*Herrmann et al., 2007*; *McGaughran et al., 2013b*). Therefore, the oriental beetle sex pheromone represents the single most ecologically relevant

semiochemical by which we could repeatedly interrogate the *P. pacificus* genes involved for host sensing and other unexplored developmental roles.

In this study, we show that although the beetle pheromone, ZTDO, is attractive to *P. pacificus* adults, the beetle host pheromone may moderate nematode infection by arresting embryonic development, inducing larval paralysis, and inhibiting dauer exit. We leveraged the ability to use exogenous cGMP to potentiate the California reference strain for ZTDO attraction to find additional genes important for ZTDO sensitivity by performing an unbiased forward genetic screen for mutants with altered responses to ZTDO. We found that a putative lipid-binding protein *Ppa*-OBI-1 mediates both beetle pheromone attraction as well as a previously unknown role of the beetle pheromone to regulate *P. pacificus* development. Our results suggest that the host pheromone may play a complex role in reducing infection or coordinating development for the dispersal cycle of the necromenic lifestyle.

## Results

### *Ppa-obi-1* mutants show altered ZTDO attraction

*P. pacificus* and *C. elegans* have diametrically opposed odor preference profiles. *Pristionchus* species are generally attracted to plant volatiles, insect pheromones, and cuticular hydrocarbons, but each *Pristionchus* species can have its own chemosensory profile reflective of its preferred beetle hosts (*Hong and Sommer, 2006a*; *McGaughran et al., 2013a*). *P. pacificus* populations found on the oriental beetle *E. orientalis* in Japan and northeastern United States are attracted to the sex pheromone of the beetle, z-7-tetradecen-2-one (ZTDO) (*Zhang et al., 1994*; *Herrmann et al., 2007*). In contrast, the compost-dwelling *C. elegans* is less attracted to or even repelled by plant and insect odors, including ZTDO, but is strongly attracted to small compounds released by microbes (*Bargmann et al., 1993*; *Hong and Sommer, 2006a*).

To identify genes required to mediate ZTDO attraction in a pheromone-insensitive reference strain, California PS312, we performed a behavior-based genetic screen for mutants that are insensitive toward ZTDO following a brief 1-hr treatment with the cell-permeable 8-br-cGMP. We isolated two Oriental Beetle pheromone Insensitive mutants, *Ppa-obi-1(tu404)* and *Ppa-obi-3(tu405)*. The chemosensory phenotype of *Ppa-obi-1* and *Ppa-obi-3* is strongly specific for ZTDO and does not affect the chemoattraction towards other insect pheromones ETDA and myristate nor plant volatiles β-caryophyllene and nicotine (*Figure 1*). Odor avoidance in *obi* mutants towards 1-octanol, a repellant common between *P. pacificus* and *C. elegans,* also resembles wild type. These results indicate that both *obi* mutants are specifically defective in sensing a beetle pheromone rather than more generally defective in chemosensation. *Ppa-obi-1(tu404)* and *Ppa-obi-3(tu405)* are non-allelic recessive mutants that complement each other in their physical and behavioral phenotypes. We proceeded to characterize *Ppa-obi-1(tu404)* based on its altered response to ZTDO attraction and its unexpected function in protecting larvae.

### ZTDO is an unsuspected potent embryocide and anesthetic of *P. pacificus* larva

Necromeny occupies a midpoint in the continuum of nematode–insect interactions that span from mutualism to pathogenicity. To address possible developmental effects of long-term exposure to ZTDO, we attempted to grow worms in the presence of the ZTDO by placing eggs onto OP50 bacteria in plates containing 10 µl of 0.5% ZTDO suspended under the plate lids, which is approximately a third of the amount used in chemotaxis assays. Surprisingly, very few of the synchronized eggs from wild type and *Ppa-obi-1(tu404)* hatched after 36 hr, compared to ~94% of the untreated eggs (*Figure 2A*). ZTDO therefore may function not just as a volatile attractant but also as an antagonistic host-derived developmental regulator.

Detailed examination revealed that *Ppa-obi-1(tu404)* eggs had significantly lower hatching rate in 0.10% ZTDO compared to wild-type PS312. Interestingly, the Washington strain that has natural cGMP-independent attraction to ZTDO was more resistant to 0.25–0.50% ZTDO than the ZTDO-insensitive California strain. Although ZTDO exposure had no noticeable effect on egg laying output by adult hermaphrodites, eggs continuously exposed to ZTDO for 2 days at an early stage remain arrested at the pre-comma globular embryonic stage, before gastrulation. The embryocidal effect of ZTDO was concentration-dependent and species-specific, since wild-type *C. elegans* eggs were completely unaffected (*Figure 2A*). *Ppa-obi-1(tu404)* hypersensitivity to the developmental and paralyzing

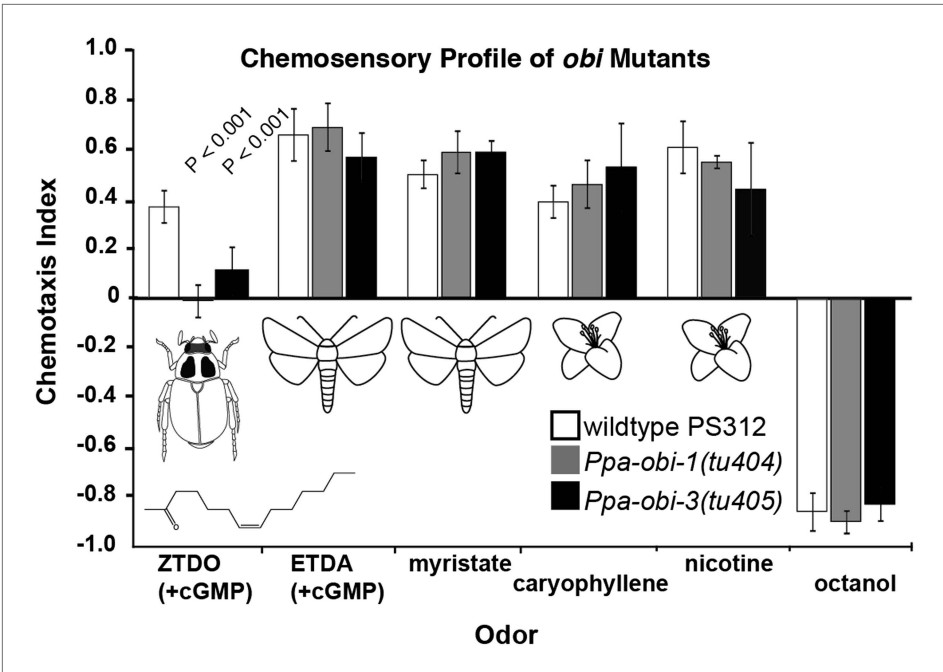

**Figure 1**. Chemosensory behavior of oriental beetle pheromone insensitive mutants. *Ppa-obi-1* and *Ppa-obi-3* mutant animals are specifically defective for sensing the host oriental beetle sex pheromone, *Z*-7-tetradece-2-one. The ZTDO structure is depicted below the beetle. In contrast, *obi* mutants' response is wild type towards another cGMP-dependent attractant, *E*-11-tetradecenyl acetate (ETDA, a lepidopteran pheromone), as well as myristate (methyl tetradecanoate, an insect allomone), plant volatiles beta-caryophyllene and nicotinic acid, and the repellent 1-octanol. N ≥ 15 replicates in ≥3 trials. Significant differences to wild type are indicated by p values (Dunnett's test) and error bars represent SEM.

effects of ZTDO also continued into the subsequent J2 and J4 larval stages (*Figure 2B,C*). Both wild-type PS312 and *Ppa-obi-1(tu404)* J2 larvae exhibited ZTDO-induced paralysis, while the Washington strain was significantly more resistant to paralysis. However, by the J4 larval stage, only *Ppa-obi-1(tu404)* was sensitive to ZTDO compared to both wild-type *P. pacificus* strains (*Figure 2D–E*). *Ppa-obi-1(tu404)* J4 larvae exposed to ZTDO displayed initial agitation, followed by full body paralysis between 90 and 120 min. In contrast, wild-type *C. elegans* J4 larvae displayed only minute sensitivity to ZTDO. To determine if the *Ppa-obi-1(tu404)* mutation renders it susceptible to paralysis from other insect pheromones, we also tested the moth pheromone ETDA and found that the J4 larvae of *Ppa-obi-1(tu404)* and the wild-type strains were not affected (*Figure 2H*). All strains, including *Ppa-obi-1(tu404)*, were insensitive to ZTDO-induced paralysis when they reach adulthood (*Figure 2F*), the dominant stage used in our chemotaxis assays. Taken together, our findings show that ZTDO is capable of arresting wild-type *P. pacificus* embryogenesis and severely paralyzing J2 larvae, indicating that ZTDO is a potential host-specific developmental regulator. We speculate that wild-type *Ppa*-OBI-1 may serve to protect *P. pacificus* embryos and reproductive larvae against the volatile anesthetic-like effects of ZTDO from the adult beetle host.

## ZTDO inhibits dauer larva exit in *P. pacificus*

The dauer larva (DL) is an alternative, non-reproductive, long-lasting stage in nematodes capable of withstanding starvation, as well as the infective stage in parasitic nematodes (*Lee, 2002*). To explore the possibility that ZTDO also acts to restrain the dispersive stage of *P. pacificus* development after hatching, we asked if ZTDO can also regulate DL exit in the presence of food using culturing plates containing 0.001–0.01% ZTDO and bacteria. Nearly all untreated wild-type *P. pacificus* DL developed into J4 larvae after 2 days on food, whereas ~40% of the wild-type worms remained as active DL on 0.01% ZTDO with OP50 food that would have otherwise promoted reproductive development (*Figure 2G*). However for *Ppa-obi-1* DL, this dauer exit defect is exacerbated by ZTDO due to

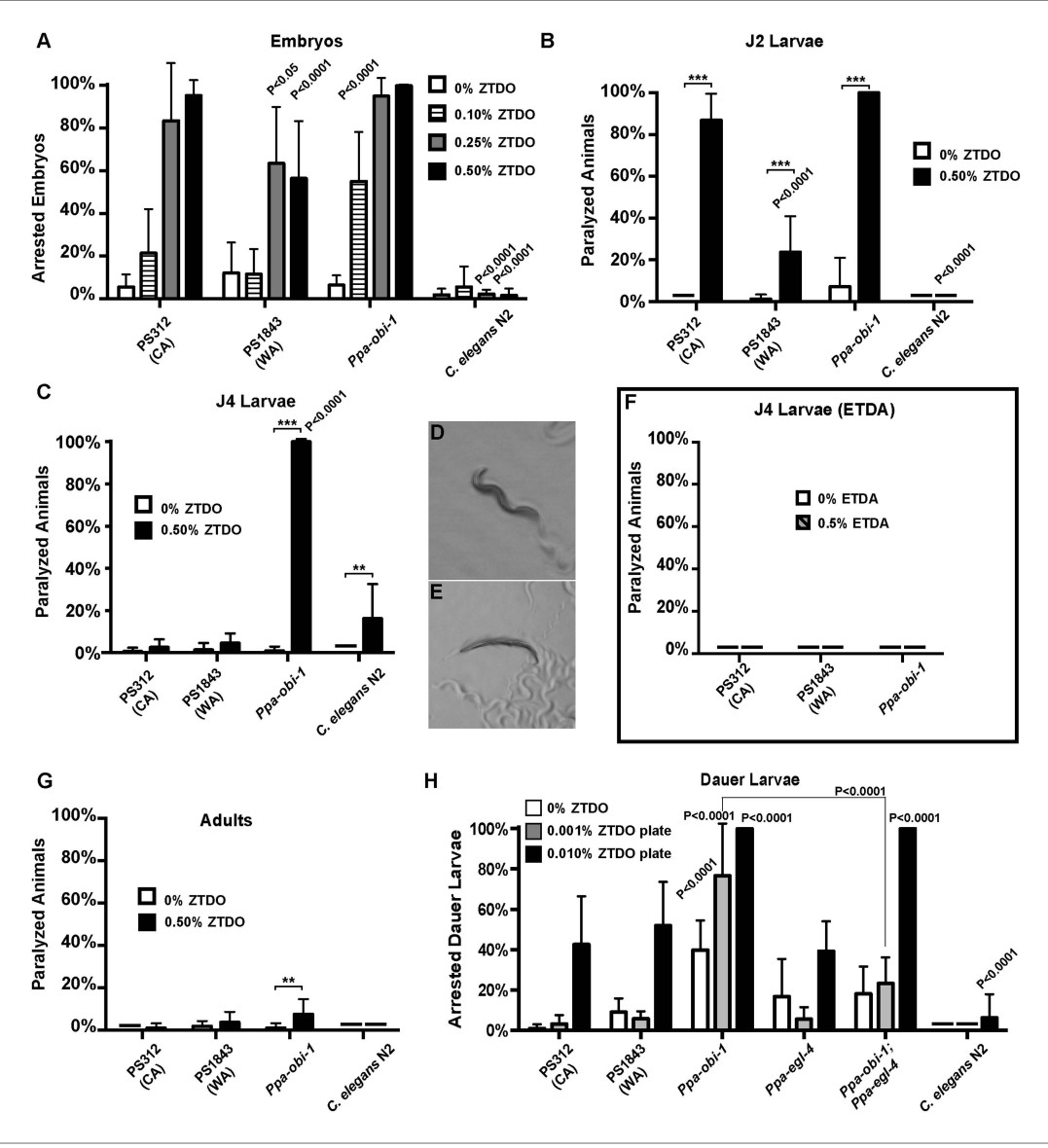

**Figure 2**. The oriental beetle pheromone ZTDO inhibits *P. pacificus* development and induces paralysis.
(**A**) *Ppa-obi-1* mutant is hypersensitive to 0.10% ZTDO compared to wild type PS312 (California). However, PS312 is more sensitive to 0.5% ZTDO than another strain, PS1843 (Washington). PS312 (n = 372, 82, 90, 320); PS1843 (n = 309, 88, 92, 237); *Ppa-obi-1* (n = 332, 69, 71, 389); *C. elegans* wild-type N2 (n = 316, 63, 84, 174). (**B**) Both wild-type PS312 and *Ppa-obi-1* mutant J2 larvae are more sensitive to ZTDO than PS1843. PS312 (n = 154, 115); PS1843 (n = 110, 102); *Ppa-obi-1* (n = 104,100); *C. elegans* wild-type N2 (n = 23, 73). (**C**) *Ppa-obi-1* J4 larvae is uniquely sensitive to ZTDO compared to PS312 (p < 0.0001). PS312 (n = 209, 290); PS1843 (n = 102, 99); *obi-1* (n = 244, 237); *C. elegans* wild-type N2 (n = 69, 109). (**D**) 90 min exposure of a *Ppa-obi-1(tu404)* J4 larvae to ethanol control compared to (**E**) 0.50% ZTDO resulted in a rigid fully paralyzed posture. (**F**) Adults are not paralyzed by ZTDO. PS312 (n = 108, 113); PS1843 (n = 114, 107); *Ppa-obi-1* (n = 109, 109); *C. elegans* wild-type N2 (n = 62, 72). (**G**) 0.01% ZTDO prevents resumption of post-dauer development in approximately 40% of wild-type animals and nearly all *Ppa-obi-1(tu404)* mutants. This ZTDO inhibition of dauer exit is dependent on the cGMP-dependent PKG, *Ppa-egl-4*. PS312 (n = 384, 654, 267); PS1843 (n = 195, 139, 101); *Ppa-obi-1* (n = 339, 343, 146); *Ppa-egl-4(tu374)* (n = 123, 181, 77); *Ppa-obi-1;Ppa-egl-4* (n = 132, 144, 63); N2 (n = 131, 109, 115). (**H**) Exposure to 0.50% ETDA moth pheromone from *Spodoptera littoralis* does not paralyze *Ppa-obi-1* mutant and wild-type J4 larvae. PS312 (n = 40, 103); PS1843 (n = 40, 64); *Ppa-obi-1* (n = 40, 193). p values indicate Dunnett's multiple comparisons test to wild-type PS312. ***p < 0.001; **p < 0.01 indicate difference between ethanol control and ZTDO exposed groups (pairwise unpaired *t* test). Error bars are SEM and *n* is the number of nematodes tested.

paralysis. Approximately 80% of the developmentally blocked *Ppa-obi-1(tu404)* DL show immobility on 0.001% ZTDO, and nearly 95% of the *Ppa-obi-1(tu404)* DL became paralyzed on 0.01% ZTDO. Although DL is the primary infective stage in many entomophilic nematodes found on live beetles (*Weller et al., 2010*), we were unable to assess DL's chemotaxis response to ZTDO because of its immobilizing effect on DL. The negative regulation of DL exit by ZTDO is specific to *P. pacificus* and barely affected *C. elegans* even at 0.01% ZTDO. These results show that 0.01% ZTDO can block active wild-type *P. pacificus* DL from progressing to the reproductive stage without causing paralysis, but 0.001–0.01% ZTDO significantly reduced dauer exit in *Ppa-obi-1* DL by primarily inducing paralysis and death.

Since *Ppa-obi-1(tu404)* has a defect in DL exit that is further enhanced by exposure to ZTDO, we asked if DL arrest in *Ppa-obi-1(tu404)* by ZTDO is similarly dependent on *Ppa*-EGL-4 function. *Egl-4* mutants are defective in a cGMP-dependent protein kinase in *C. elegans,* which affect dauer formation as well as chemosensation (*Daniels et al., 2000*; *L'Etoile et al., 2002*). We found that while the *Ppa-egl-4(tu374)* loss-of-function mutation did not significantly affect the ability to exit the dauer stage, the *Ppa-egl-4(tu374)* mutation in the *Ppa-obi-1(tu404)* background did partially suppress the inhibitory effect of ZTDO, such that *Ppa-egl-4;Ppa-obi-1* post-dauer development on 0.001% ZTDO is higher than *Ppa-obi-1(tu404)* (*Figure 2G*). The *Ppa-egl-4(tu374)* mutation did not, however, reduce the ZTDO inhibition of DL exit in the presence of a higher amount of 0.01% ZTDO. This outcome suggests that the susceptibility of *Ppa-obi-1(tu404)* DL to ZTDO is partially dependent on *Ppa-egl-4*. While the loss of *Ppa-obi-1* alone can cause reduced rate of DL exit on food independent of ZTDO, the aggravated DL arrest by ZTDO in *Ppa-obi-1(tu404)* is partially mediated by Ppa-*egl-4.*

Although wild-type *P. pacificus* adult animals are attracted to ZTDO, the host beetle sex pheromone regulates *P. pacificus* development by inhibiting embryogenesis, restricting DL exit, and inducing paralysis in larval stages. This ZTDO sensitivity is the most acute at the embryonic stage and is likely to be buffered by wild-type *Ppa*-OBI-1 in later developmental stages as the strongest hypersensitivity to ZTDO was observed in the J4 stage of the *Ppa-obi-1(tu404)* mutant. By contrast, the PS1843 Washington strain that is naturally attracted to ZTDO is more resistant to ZTDO in non-dauer stages compared to the PS312 California wild type. Further studies in other natural populations are needed to determine the correlation between ZTDO attraction and resistance to ZTDO-induced paralysis in *P. pacificus*.

## Molecular cloning of *Ppa-obi-1(tu404)*

In addition to the specific chemosensory defect in *Ppa-obi-1(tu404)* mutants, we also noticed a linked phenotype in its body morphology. Detailed measurements indicate that *Ppa-obi-1(tu404)* animals are significantly longer and thinner than wild type (*Figure 3*), resulting in a slim body phenotype with adult *Ppa-obi-1(tu404)* hermaphrodites ~20% longer than wild-type animals. The slim body phenotype may also contribute to *Ppa-obi-1(tu404)'s* moderate egg-laying defect (*egl*; *Table 1*) and reduction in hatching rate similar to *Ppa-egl-4(tu374)* mutants, possibly due to defects in vulva formation and cuticle development. Using molecular markers and the slim phenotype, we were able to delineate the *Ppa-obi-1* lesion to an ~138 kb region on Supercontig 1 of chromosome I between markers SSLP21 and SSLP17 (*Figure 4A*). We subsequently identified all polymorphic sites between PS312 and *Ppa-obi-1(tu404)* in the fine mapping interval by Illumina whole genome sequencing and focused on disruptions of predicted open reading frames. We found a single base pair deletion in a predicted coding region orthologous to *C06G1.1* in *C. elegans* and confirmed this *tu404* deletion by PCR amplification and Sanger sequencing of both wild type and *Ppa-obi-1(tu404)* genomic and cDNA. The *tu404* allele contains a single base pair deletion in exon 9. The resulting frame shift and nonsense mutation are predicted to stop translation half way into the reading frame to result in a hypomorphic allele. In the two *Ppa-obi-1* splice forms, the longer transcript contains a VKDDDPR peptide that may be the result of an alternative splice donor site. Both splice variants were isolated from the California and Washington strains in equal abundance by sub-cloning RT-PCR products (data not shown), and no differences in nucleotide sequence were found in the coding region of these two strains. Therefore, splice form variation in *Ppa*-OBI-1 is unlikely to contribute to the difference in insect pheromone attraction between these two strains. The region encoding for VKDDDPR is also absent in all the *Ppa-obi-1* orthologs we have examined in available databases (*Figure 4—figure supplement 1*).

## Transgenic rescue and comparison of lipid-binding proteins

To confirm that *Ppa-obi-1* is indeed responsible for the lack of ZTDO attraction in adults and for hypersensitivity to ZTDO-induced paralysis in the J4 larvae, we made transgenic lines containing a 4.3 kb

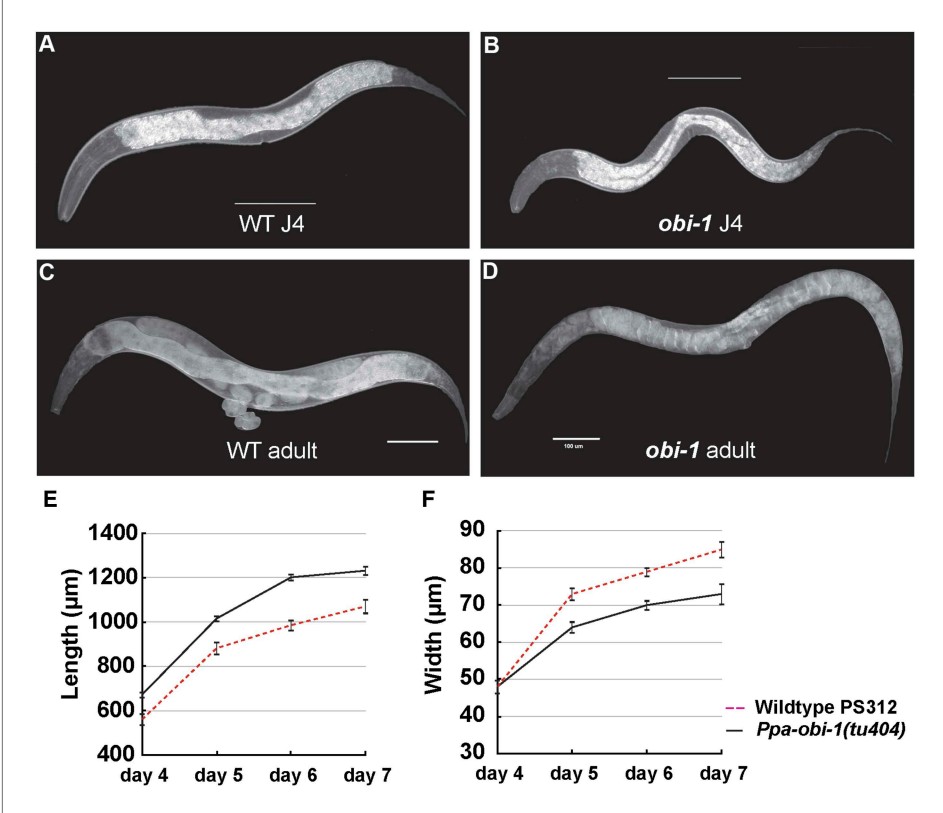

**Figure 3**. *Ppa-obi-1* mutants are long and slim. (**A** and **B**) Wild-type J4 larva and young adult. (**C** and **D**) *Ppa-obi-1* J4 larva and young adult. (**E** and **F**) The body lengths and widths of wild-type (dotted red line) versus *Ppa-obi-1* mutants (solid black line) recorded over 4 days post J4 stage (μm); n = 15–30 animals for each time point. Error bars are SEM. Scale bars denote 100 μm.

*Ppa-obi-1* promoter driving the *Ppa-obi-1* full-length cDNA (*Schlager et al., 2009*). The introduction of the wild-type *Ppa-obi-1* transgene restored the cGMP-dependent ZTDO attraction in *Ppa-obi-1(tu404)* adults compared to *Ppa-obi-1(tu404)* adults and sibling animals that did not segregate for the rescue construct (*Figure 4C*). Furthermore, the *Ppa-obi-1* rescue transgene significantly shortened the body length of *Ppa-obi-1(tu404)* adults (*Figure 4D*), confirming that both the ZTDO chemosensory and the long body length phenotype used for subsequent positional mapping were due to the same mutation in *Ppa-obi-1(tu404)*. Lastly, we tested the ability of the wild-type *Ppa-obi-1* transgene to reduce the ZTDO hypersensitivity and found that it significantly reduced the percentage of ZTDO-induced paralysis in *Ppa-obi-1(tu404)* J4 larvae (*Figure 4E*). *Ppa-obi-1* therefore mediates ZTDO chemosensation, body dimensions, and ZTDO susceptibility.

Sequence analyses of coding regions show that *Ppa-obi-1* and *C06G1.1* encode for proteins that represent a class of highly conserved but functionally diverse group of secreted lipid-transfer/lipid-binding proteins present in almost all metazoans except *Drosophila* (*Beamer et al., 1998*), including lipopolysaccharide-binding proteins (LBP) (*Tobias et al., 1997*), bactericidal permeability-increasing proteins (BPI) (*Weiss et al., 1978*), phospholipid transfer proteins (PLTP) (*Gautier and Lagrost, 2011*), and cholesteryl ester transfer proteins in mammals (CETP) (*Kawano et al., 2000*). The LBP/BPI/CETP/PTLP family in humans is known to play a role in immunological responses to bacterial pathogens and mediate the transfer of cholesteryl ester and triglycerides in the blood plasma

**Table 1.** *Ppa-obi-1* affects hatching rate and brood size

|  | Hatching rate (%) | Brood size (3 days) | n |
|---|---|---|---|
| Wild-type(PS312) | 95 ± 3 | 156 ± 16 | 18 |
| *Ppa-obi-1(tu404)* | 85 ± 3 | 103 ± 17 | 15 |

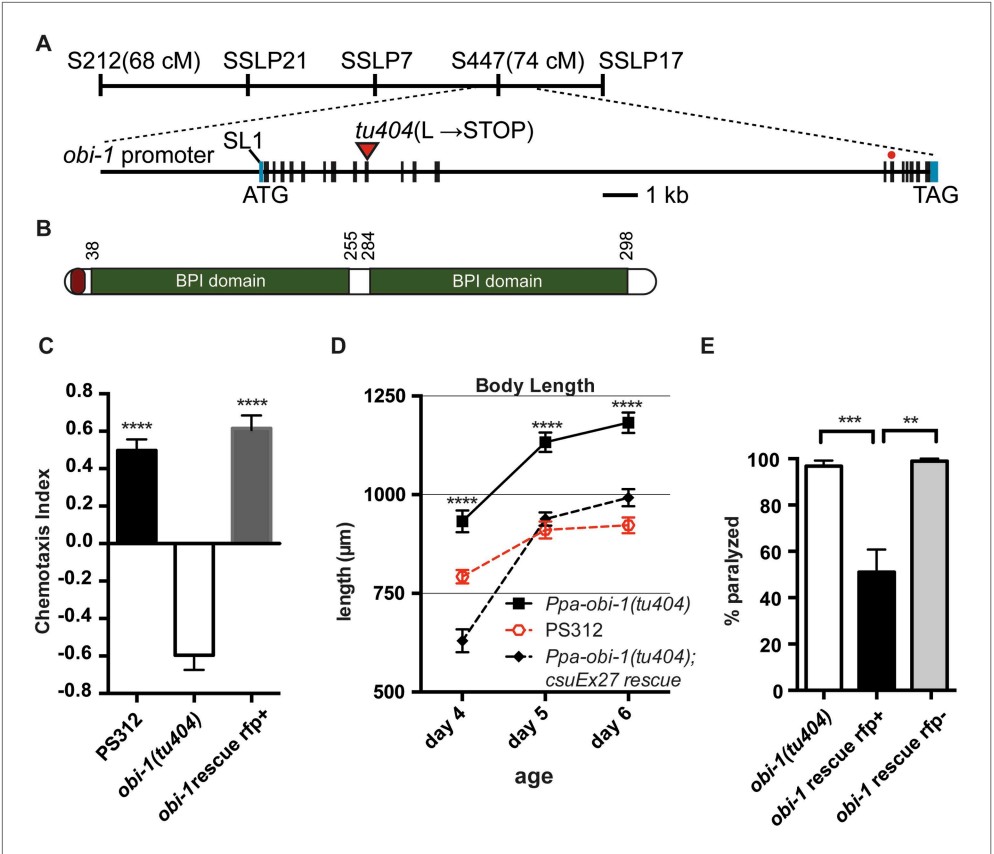

**Figure 4**. *Ppa-obi-1* gene structure and protein comparisons. (**A**) The order of mapping markers is depicted on top (not to scale). The *Ppa-obi-1* mRNA is SL1 trans-spliced and contains 19 exons spanning ~19 kb. The 50 bp 5' UTR and 341 bp 3' UTR are indicated in light blue. The *tu404* allele contains a single base pair deletion in exon 9 that results in a frameshift, converting a codon to a stop codon (inverted triangles). The red dot represents the splice variant. (**B**) A diagram of the secretion and two BPI lipid-binding domains (PFAM and SMART). (**C–E**) Transgenic rescue of the *Ppa-obi-1(tu404)*. (**C**) A 4.3 kb promoter driving *Ppa-obi-1* cDNA completely rescued cGMP-dependent ZTDO attraction in *Ppa-obi-1(tu404)* mutants. PS312 (N = 21); *obi-1* (N = 15); *obi-1* rescue (N = 12). The *Ppa-obi-1* transgenic rescue (**D**) restored the body length defect in *Ppa-obi-1(tu404)* mutants, PS312 (n = 15); *obi-1* (n = 15); *obi-1* rescue (n = 19) as well as (**E**) significantly reduced ZTDO-induced paralysis in *Ppa-obi-1(tu404)* J4 larvae. PS312 (N = 11); *obi-1* rescue *rfp*+ (N = 12); *obi-1* rescue *rfp*− (N = 5). **p < 0.01; ***p < 0.001; ****p < 0.0001 (Tukey's multiple comparisons test). Error bars represent SEM. N is the number of assays performed and *n* is the number of individual worms scored for each time point.

The following figure supplements are available for figure 4:

**Figure supplement 1**. An alignment of putative *Ppa*-OBI-1 protein orthologs.

**Figure supplement 2**. A maximum likelihood relationship of lipid-binding proteins in *P. pacificus* and *C. elegans*.

**Figure supplement 3**. Chemosensory behavior of *C. elegans* mutants *C06G1.1* and *nrf-5*.

**Figure supplement 4**. Reduction of C06G1.1 function produce a slimmer body phenotype.

---

(*Bruce et al., 1998*; *Kawano et al., 2000*; *Schultz and Weiss, 2007*). *Ppa*-OBI-1 and C06G1.1 each contain two predicted lipid-binding domains as well as a predicted secretion peptide sequence in the N-terminal regions (PFAM and SMART; SignalP 4.0 Server) (*Figure 4—figure supplement 1*). To identify all members of the LBP family in the sequenced genome and transcriptome of *P. pacificus*, we performed reciprocal BLASTP searches (WormBase WS221), which indicated 12 members predicted to be encoded in the *P. pacificus* genome. Maximum likelihood phylogenetic analysis of the OBI-1/C06G1.1

LBP protein family indicates 10 closely related orthologous pairs in the two species. Interestingly, the lengths of the branches suggest that all the orthologs between *P. pacificus* and *C. elegans* share similar divergence times, with the exception of the OBI-1/C06G1.1 pair that seemed to have undergone faster divergence relative to other paralogs (*Figure 4—figure supplement 2*). Consistent with their roles in immune response in mammals, a member of the LBP family in *C. elegans*, F44A2.3, was found to be up-regulated in response to several species of pathogenic bacteria (*Wong et al., 2007*). Another *C. elegans* LBP member, NRF-5, is partially necessary for nose contraction in response to the anti-depressant fluoxetine (Prozac) (*Choy et al., 2006*), and for mediating phosphatidylserine transfer from apoptotic cells to engulfing cells during cell corpse removal (*Zhang et al., 2012*). We found that *nrf-5(sa513)* mutants are also defective in attraction to 2,3-butanedione (diacetyl), and isoamyl alcohol, although the *C06G1.1(vc20357)* allele with a missense mutation did not show chemosensory defects (*Figure 4—figure supplement 3*). Thus LBP members may have a conserved previously uncharacter-ized role in chemosensation in diverse nematodes.

To determine if *C06G1.1* can also regulate body morphology in *C. elegans*, we performed *C06G1.1* RNAi by feeding. *C06G1.1* RNAi was previously observed to generate a long body phenotype in a large-scale RNAi screen in *C. elegans* (*Simmer et al., 2003*). We used this fragment (Fragment 1), along with another *C06G1.1* fragment more proximal to the 3′ end (Fragment 2), to quantify the effect of *C06G1.1* RNAi in the enhanced RNAi *rrf-3(pk1426)* strain. Although *C06G1.1* RNAi did not signifi-cantly change the absolute body length of adult worms, we found RNAi against *C06G1.1* using both fragments resulted in a significantly slimmer body proportion (length/width ratio) similar to the *Ppa-obi-1(tu404)* mutant (*Figure 4—figure supplement 4*). A missense mutation in *C06G1.1(VC20357)* also resulted in a similar slim body phenotype. Thus both *Ppa*-OBI-1 and its *C. elegans* ortholog also share a conserved function in regulating body proportions.

## *Ppa-obi-1* is expressed in cells with putative roles in secretion

To characterize which tissue types Ppa-OBI-1 functions in, we sought to determine the *Ppa-obi-1* expres-sion pattern using a *Ppa-obi-1p::gfp* transcriptional reporter gene containing a 4.3 kb promoter frag-ment upstream of the *Ppa-obi-1* ATG initiation codon. Our analysis was based on a stably integrated transgenic line *csuIs01* and parsimonious inferences to homologous *C. elegans* cell positions and known cellular functions. Detailed analyses of all developmental stages show that the earliest *Ppa-obi-1* expres-sion occurs in at least one cell in the pre-comma embryonic stage but soon disappears until weak expression reappears in the hyp7 and seam cells in J4 larvae (*Figure 5*). The most robust and diverse *Ppa-obi-1* expression was found in mid-J4 hermaphrodites, which is consistent with real-time quanti-tative PCR results indicating that transcript levels in J2 and J3 larvae are less than in J4 larvae (data not shown). In the mid-J4 stage, *Ppa-obi-1* is strongly expressed in the major hypodermal syncytium and the lateral hypodermal cells that cover most of the body (hyp7 and seam cells) as well as the ventral tail (hyp 8 and 9) (*Figure 5*). In contrast to young adult *C. elegans,* which have 16 seam cells on each lateral surface (H0–H3, V1–V6, T descendants; The WormAtlas), we counted only 15 seam cells per side in *P. pacificus* J4 larvae (n = 20). *Ppa-obi-1* is also strongly expressed in the P(5–7).p cell descend-ants that form the vulva lumen (vulC, D, E) during the mid-J4 stage until just prior to vulval eversion as well as in the vulva muscles (likely vm1, 2) (*Figure 5B* inset; *Figure 5—figure supplement 1*).

Most notably, *Ppa-obi-1* expression was observed in the two bilaterally symmetrical specialized epi-thelial cells that envelope the chemosensory neurons, known as the amphid sheath cells (*Figure 5A–B*, arrows). The amphid sheath cells are support cells for the sensory amphid neurons and completely enclose the ciliated endings of the AWA, AWB, and AWC chemosensory neurons in *C. elegans*. *Ppa-obi-1p::gfp* expression in the putative amphid sheath cells lie dorsal–lateral and posterior to the DiI-labeled amphid neurons ASK, ADL, and ASI (*Figure 5C*). DiI staining of amphid neurons in *Ppa-obi-1* mutants is superficially wild type, suggesting that *Ppa*-OBI-1 is unlikely to be involved in the develop-ment of the amphid neurons (data not shown). We also noticed expression in four unknown cell types anterior to the metacarpus (anterior pharyngeal bulb), which could be putative neuronal support cells for the inner labial or outer labial quadrant sensilla (IL or OLQ) (*Figure 5—figure supplement 1*). *Ppa-obi-1* is also expressed in putative duct and excretory cells on the ventral anterior side that form part of the excretory system (*Figure 5B*, arrowhead). Based on conserved DiI labeling of *P. pacificus* amphid neurons and cell positions, the *Ppa-obi-1* expression profile and secretion peptide sequence sug-gest that *Ppa*-OBI-1 is likely an extracellular protein involved in sensory neuron support, epithelial cell development, and osmoregulation.

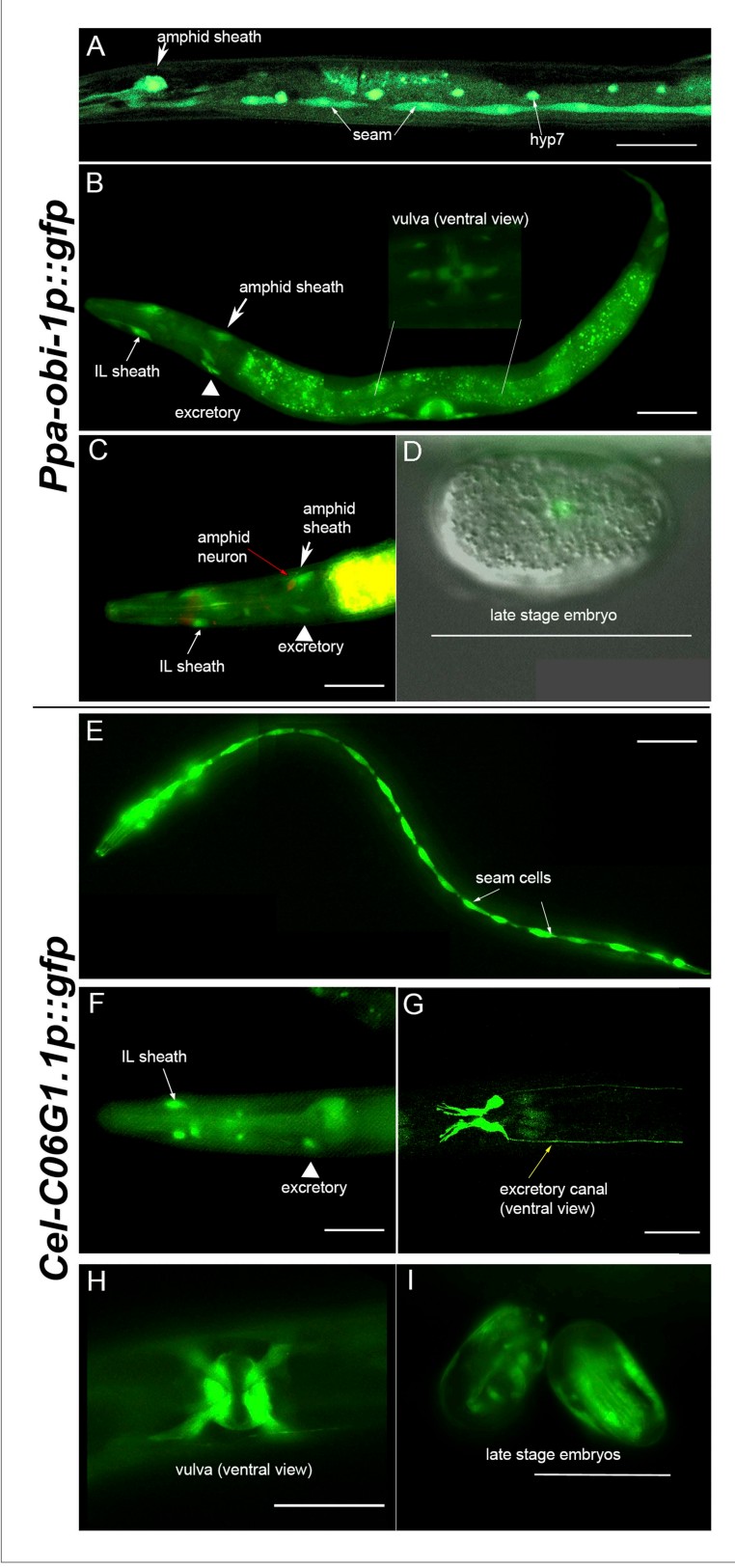

**Figure 5**. *Ppa-obi-1 and C06G1.1* expression in their respective nematode species. (**A**–**D**) An integrated *Ppa-obi-1* promoter::*gfp* transgene (*csuIs01*) in *P. pacificus*. (**A**) Lateral view of a single animal showing prominent expression in an amphid sheath cell, seam cells, and the syncytial hypodermal cell (hyp7) in a J4 hermaphrodite. (**B**) Medial
*Figure 5. Continued on next page*

*Figure 5. Continued*

view of a transgenic animal with expression in the putative dorsal and ventral inner labial (IL) or outer labial (OL) neuronal support cells (socket or sheath), the duct and excretory cells on the ventral side, the amphid sheath cell on the dorsal side (arrow head), and the vulval muscles. The inset shows a ventral view of the young adult vulva in another animal. Autofluorescence is visible in the intestine. (**C**) A DiI-labeled neuron (red arrow) anterior to a *Ppa-obi-1p::gfp* expressing amphid sheath cell (white arrow) is dorsal to an excretory cell (arrow head). A putative IL or OL socket or sheath cell (arrow) is also visible. (**D**) DIC and fluorescent overlay image of expression in early *P. pacificus* embryo. (**E**–**I**) *C06G1.1* promoter::*gfp* transgenes (*csuEx02, csuEx03*) in *C. elegans*. (**E**) Lateral view of a seam cell expression in a L3 hermaphrodite. (**F**) Medial view of putative inner or outer labial neuron support cells (arrow) and excretory cells (arrowhead). No amphid sheath shows gfp expression. (**G**) Ventral view of head region shows prominent expression in excretory cells and canal (arrow). (**H**) Ventral view of young adult vulva. (**I**) Embryonic expression in *C. elegans* embryos appears later, stronger, and in more cells than in *Ppa-obi-1*. Anterior is left and the scale bars represent 50 µm. Each panel is representative of ≥50 animals.

The following figure supplements are available for figure 5:

**Figure supplement 1**. *Ppa-obi-1* is expressed in putative amphid socket cells, amphid sheath cells, and vulval cells.

**Figure supplement 2**. Targeted misexpression of *Ppa-obi-1* in *C. elegans*.

## *Ppa-obi-1* but not *C06G1.1* is expressed in glial-like sheath cells

Although *Ppa*-OBI-1 and its ortholog C06G1.1 in *C. elegans* share 61% amino acid identity, *P. pacificus* and *C. elegans* have opposite responses to the pheromone ZTDO. While small changes in sequence can lead to large changes in protein function, the absence of expression of the *C. elegans obi-1* ortholog in amphid sheath cells suggests that changes in gene expression pattern may also play a role in determining ZTDO sensitivity. We therefore sought to determine if the *Ppa-obi-1* ortholog *C06G1.1* have shared tissue expression with *Ppa-obi-1*. We examined endogenous *C06G1.1p::gfp* expression in *C. elegans* in five independent transgenic lines with extrachromosomal arrays and found strong mid-L4 larvae expression in the seam cells, two tail hypodermal cells, four cells in the excretory duct system (duct, excretory, gland cells), vulval muscles, and possible IL or OLQ sheath or socket neuronal support cells similar to the expression pattern observed for *Ppa-obi-1* (*Figure 5E–H*). In contrast to the earliest *Ppa-obi-1* expression in pre-comma stage, *C06G1.1* expression is detectable during late embryogenesis in putative seam cells (*Figure 5I*). This seam cell expression is earlier than *Ppa-obi-1p::gfp* in the L1 stage, and is maintained until the young adult stage. A reason for this earlier seam cell expression could be that the equivalent *P. pacificus* J1 larvae remain inside the egg while *C. elegans* eggs hatch as L1 larvae (*Fürst von Lieven, 2005*). Like *Ppa-obi-1*, *C06G1.1* expression is largely absent in all tissue types when the transgenic animals become 1-day old adult hermaphrodites, coinciding with the fusion of seam cells into a syncytium. This observation indicates that *C06G1.1* expression in the excretory and epithelial system may be coordinated and involved in seam cell development.

Our expression analysis of *obi-1* orthologs in the their respective nematode species revealed that the most striking difference is the lack of expression in the amphid sheath cells in *C. elegans*, suggesting that unlike *Ppa-obi-1*, *C06G1.1* is not likely to function directly with chemosensory neurons in *C. elegans*. In addition to differences in gene expression pattern, the possibility that there is a ZTDO receptor that interacts with *Ppa*-OBI-1 in *P. pacificus* but not in *C. elegans* may also help to explain how *Ppa-obi-1* orthologues confer different ZTDO sensitivities in the two species during certain developmental stages. To explore yet another possibility that changes in both protein function and expression may lead to differences in ZTDO sensitivity, we expressed the *Ppa-obi-1:cDNA* under the *C. elegans* amphid sheath cell-specific promoters *T02B11.3* and *F16F9.3* in *C. elegans* (*Bacaj et al., 2008*; *Kramer et al., 2010*; *Oikonomou et al., 2011*; *Procko et al., 2011*). Both *T02B11.3p::Ppa-obi-1:gfp(csuEx29)* and *F16F9.3p::Ppa-obi-1:gfp(csuEx30)* transgenic animals showed strong GFP expression in the amphid sheath cells (*Figure 5—figure supplement 2*) as well as phasmid sheath cells in the tail region (not shown). While the targeted misexpression of *Ppa-obi-1* in *C. elegans* did not alter the strong avoidance behavior towards either 2% or 10% ZTDO, *Ppa-obi-1* expression in *C. elegans* did reduce the paralytic effects of ZTDO (*Figure 5—figure supplement 2*). Because wild-type *C. elegans* N2 is not susceptible to ZTDO-induced paralysis at 0.5% ZTDO that *P. pacificus* is, we increased the stringency

of the assay by using 10% ZTDO, incubating for 16 hr, and exposing the animals in the absence of OP50 food. Under this enhanced ZTDO paralysis assay, we found that *F16F9.3p::Ppa-obi-1:gfp(csuEx30)* transgenic L4 larvae but not *T02B11.3p::Ppa-obi-1:gfp(csuEx29)* exhibited a small but statistically significant increase in protection from ZTDO-induced paralysis compared to wild-type N2 (*Figure 5—figure supplement 2C,D*). Because we could not detect apparent differences in GFP expression patterns between the two amphid promoters, the ability of *F16F9.3p::Ppa-obi-1:gfp* to reduce ZTDO paralysis may be due to subtle transient expression differences. These results suggest that while the expression of the *Ppa*-OBI-1 protein in the *C. elegans* amphid sheath cells is not sufficient to reduce avoidance to ZTDO, it can contribute to the *obi-1*-dependent protection from ZTDO-induced paralysis.

## Discussion

Every organismal interaction is a legacy of its evolutionary arms race. We have identified an important and likely co-evolving trait in the interaction between *P. pacifcus* nematodes and its beetle hosts. Although not symbiotic, necromenic associations between nematodes and insects have been considered commensalistic or mutualistic, but our results suggest antagonistic interactions between the larval stages of *P. pacificus* and the adult oriental beetle hosts that produce the ZTDO sex pheromone. Attraction and resistance to ZTDO in adult *P. pacificus* further points to a refined attenuated antagonism in their co-evolutionary relationship (*Thompson, 2005*). Our data support the model that *Ppa*-OBI-1 is required to protect *P. pacificus* larvae against a host beetle pheromone ZTDO as well as to enable *P. pacificus* attraction to ZTDO. Furthermore, two different geographical isolates and developmental stages have varied responses to the oriental beetle pheromone. ZTDO may be a species-specific volatile synomone that acts through a lipid-binding protein to moderate nematode infection by arresting embryonic development, inducing larval paralysis, and inhibiting dauer exit. To the best of our knowledge, this is the first genetically defined reception to a species-specific compound mediating the interaction between a nematode and its insect host.

### The role of *Ppa*-OBI-1 in ZTDO sensitivity

Sensitivity to ZTDO-induced paralysis/arrest is *Ppa-obi-1*-dependent predominantly during the J4 larval stage and, to a lesser degree, in the embryos and dauers. Given the diversity of tissues expressing *Ppa-obi-1* in the J4 larvae, it is not apparent which tissues require *Ppa-obi-1* expression for proper chemosensation and if the same tissues are involved with protection from ZTDO-induced paralysis in wild-type animals (*Figure 6*). Our gene expression studies show that despite coding highly similar proteins, *P. pacificus Ppa-obi-1* is uniquely expressed in the amphid sheath cells compared to its *C. elegans* ortholog *C06G1.1*. Based on this species-specific expression pattern, we are tempted to speculate that these chemosensory glial cells, the IL and amphid sheath cells, play a dominant role in mediating ZTDO sensitivity. However, additional experiments will be needed to test the possibility that epidermal cells such as the vulva, hypodermis, and seam cells, or cells in the excretory system may also mediate ZTDO sensing.

Given the growing evidence that the amphid sheath cells in *C. elegans* are critical for proper neuronal development and function of the 12 pairs of chemosensory amphid neurons (*Shaham, 2006*), we propose the following models on how the *Ppa*-OBI-1 protein can act in the sheath cells to directly or indirectly modulate ZTDO chemosensation. In *C. elegans*, many proteins have enriched expression in the amphid sheath, such as DAF-6 and the secreted venom allergen-like VAP-1, and these secretions are corroborated by dense populations of vesicles in amphid sheath cells under electron microscopy (*Bacaj et al., 2008*; *Doroquez et al., 2014*). However, it is not yet known which amphid proteins, if any, are required in processing odors using behavioral assays. Interestingly, DAF-6 is a Patched-related protein with expression pattern that overlaps with *Ppa-obi-1* expression (amphid lumen, excretory canal, vulva lumen) and is required for chemotaxis and dauer formation (*Perens and Shaham, 2005*). Similarly, *Ppa-obi-1* expression in the putative sheath or socket cells of the inner labial neurons may also be involved in ZTDO sensing, as recent ultrastructural electron micrograph reconstructions indicate that BAG (gas sensing) and FLP (nociceptive) neurons in *C. elegans* associate closely with the socket cells of the inner labial sensillae (*Doroquez et al., 2014*). One possible mode of action for ZTDO may be its ability to change cell membrane permeability, thereby indirectly affecting the functions of the glia-associated neurons. Hence, *Ppa*-OBI-1 may function to counter the cell membrane permeability changes brought about by ZTDO by interacting with cell membrane lipids. Alternatively, *Ppa*-OBI-1 could mediate pheromone binding directly. In lepidopterans such as the wild silk moth

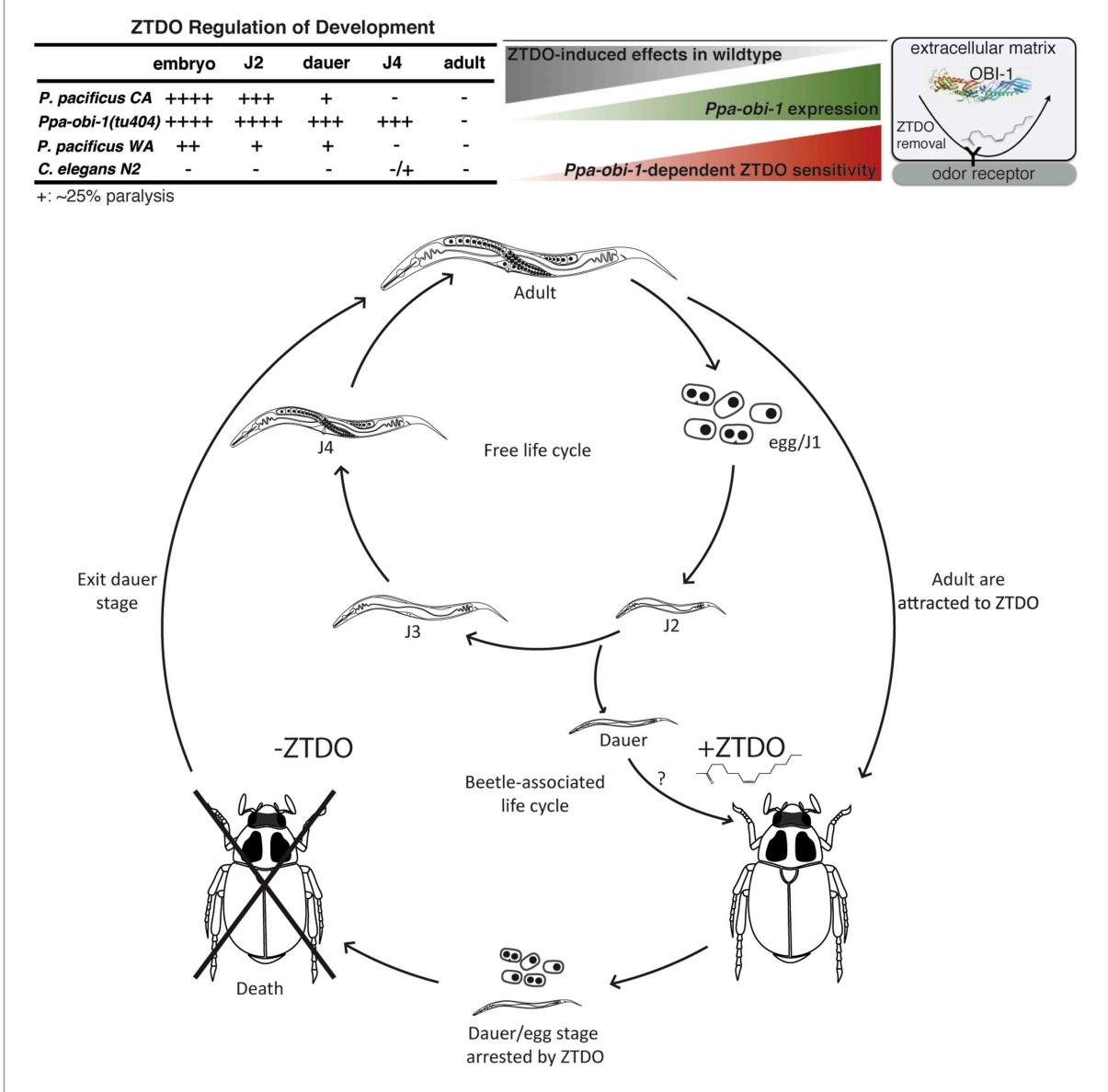

**Figure 6**. Summary and model of the oriental beetle pheromone ZTDO on *P. pacificus* development and lifecycle. (Top) Peak *Ppa-obi-1* expression in the wild-type J4 larvae coincides with the least sensitive developmental stage to the paralytic effects of the beetle pheromone ZTDO (at 0.5%). *Ppa-obi-1* mutants are most sensitive to ZTDO-induced paralysis in the J4 stage. Our working model on *Ppa*-OBI-1 functions in the J4 larvae hypothesizes that *Ppa*-OBI-1 proteins in the extracellular space, such as the lumen of the amphid sheath cell, may be required to remove ZTDO bound to odor receptors on neurons. (Bottom) This working model shows both *P. pacificus* adults and dauer larvae from non-beetle populations associate with the host oriental beetle through attraction to the beetle's sex pheromone ZTDO, whereby the progeny born on the beetle are arrested as eggs or dauer larvae. Death of the host allows microorganism to grow, which then encourages dauer larvae to exit and resume reproductive development. Several iterations of asynchronous nematode populations likely complete each dispersal cycle.

*Antheraea polyphemus* and the cotton leafworm *Spodoptera littoralis*, pheromone-degrading and odorant-degrading enzymes (PDEs and ODEs) are esterases specifically expressed in the male antennal sensillar lymph analogous to the amphid lumen (***Ishida and Leal, 2005***; ***Durand et al., 2011***). The termination of the signal by removing odors that have already triggered a response is an important process for preventing odor adaptation and increasing odor sensitivity. Thus, it is possible that *Ppa*-OBI-1 acts as a solubilizing factor for the lipid pheromones in the excretion-filled amphid sheath lumen (***Figure 6***). The weak ability of amphid sheath-specific expression of *Ppa*-OBI-1 to confer ZTDO

protection in *C. elegans* suggests that both expression pattern and protein function could be important factors in ZTDO sensing in *P. pacificus*. Future work will be necessary to determine if *Ppa-obi-1* expression restricted to the amphid sheath is sufficient to rescue the ZTDO attraction defect in *P. pacificus* and if the amphid sheath cells are required to mediate ZTDO sensing.

### The role of *Ppa*-OBI-1 in neutralizing a host nematocide

Given that *Ppa-obi-1*-independent susceptibility to ZTDO occurs during the J2 larval stage when *Ppa-obi-1* is expressed in the seam and hypodermal cells rather than in the amphid sheath that is only active in the J4 larvae, ZTDO-induced paralysis likely acts through the extracellular matrix of the nematode cuticle. *P. pacificus* adults, whose cuticle may be compositionally different from the larval cuticle, are not susceptible to ZTDO. However in *P. pacificus* embryos, the susceptibility of ZTDO may be due to ZTDO limiting gas or ion exchange with the environment rather than direct interaction between ZTDO and cellular components, given that nematode egg shells are considered to have very limited permeability accessible only to small molecular weight solutes and gas molecules (*Chitwood and Chitwood, 1974*). Since ZTDO exposure does not lead to embryonic arrest and paralysis in *C. elegans*, ZTDO must also interact with other non-conserved *P. pacificus*-specific molecules that have yet to be identified. There are very few known nematocides from natural compounds. One rare example comes from the oyster mushroom *Pleurotus ostreatus,* which secretes droplets that can immobilize rhabditid nematodes in an hour, though its effects on embryos have not been investigated (*Barron and Thorn, 1987*; *Stadler et al., 1994*). Future work on the different susceptibility to ZTDO may help to answer if there is a wider pattern of negative correlation between ZTDO attraction and ZTDO-induced paralysis in other natural isolates, as observed in the California and Washington strains. Such antagonistic traits may reveal possible co-evolution between host finding and host tolerance in necromenic associations.

We further speculate that in the natural ecology of *P. pacificus,* ZTDO may enable the beetle host to limit the nematode number and to thus synchronize the developmental stage of infective *P. pacificus* by arresting embryogenesis and lethally paralyzing J2 larvae. The unexpected role of a *P. pacificus* attractant to act as a developmental regulator against the younger larval stages, particularly the dauer larvae (DL), suggests that the ZTDO-resistant adult stage is a potential dispersal stage. This attenuated antagonistic interaction also revealed that nematode larvae and adults may be susceptible to distinct selective pressures. *P. pacificus* DL does not disperse readily, likely due to the secretion of the wax ester nematoil that enables *P. pacificus* to form dauer towers when potential hosts are nearby (*Penkov et al., 2014*). Therefore under natural conditions, lower concentrations of ZTDO than the concentrations we have tested may serve as a close-range host dwelling strategy by inhibiting DL dispersal to form dauer aggregates. The developmental role of ZTDO may be part of a mechanism to limit nematode dispersal until the beetles mature. In these respects, the promotion of a non-DL dispersal stage and inhibition of DL dispersal by ZTDO resemble other known host-derived compounds found in the entomophilic nematode *B. xylophilus* and the phoretic nematode *C. japonica* (*Okumura et al., 2013*; *Zhao et al., 2013*). Attraction and paralysis by ZTDO may have co-evolved in the necromenic association to allow successive populations of founder *P. pacificus* DL and adults to infect beetles, culminating in arrested DL remaining on sexually mature beetles during the dispersal cycle. The pheromone arrested DL state may also help to account the observation that *Pristionchus* nematodes emerge several days later than rhabditids from the same species of host beetles. Our findings that a lipid-binding protein has a role in coordinating chemosensation, body proportions, and protection from a host pheromone encourages fresh inquiries on additional genes involved in ZTDO sensory signaling and its mechanism of action.

## Materials and methods

### Strains

All nematodes were raised on NGM media containing OP50 *E. coli* at ~23°C as previously described (*Hong and Sommer, 2006a*; *Hong et al., 2008a*). The following strains were used: ***P. pacificus:*** California (PS312) wild type, Washington (PS1843), *Ppa-obi-1(tu404)*I; *Ppa-egl-4(tu374)*IV (null); *Ppa-obi-3(tu405)*; *csuIs01* [*Ppa-obi-1p::gfp; Ppa-egl-20p::rfp; PS312 gDNA*]X; *rlh49(csuIs01; tu374)* '*Ppa-obi-1p::gfp; Ppa-egl-4*'; *rlh50(csuIs01; tu92)* '*Ppa-obi-1p::gfp; roller*'; *rlh51(csuIs01; tu404)* '*Ppa-obi-1p::gfp; Ppa-obi-1*'. ***C. elegans:*** Bristol(N2) wild type, *nrf-5(sa513)*, *csuEx02* [*C06G1.1p::gfp; Cel-myo-2::DsRED*];

*csuEx03* [*C06G1.1p::gfp; pRF4 roller*], *csuEx29* [*T02B11.3p::Ppa-obi-1:gfp; Cel-myo-2::DsRED*], and *csuEx30* [*F16F9.3p::Ppa-obi-1:gfp; Cel-myo-2::DsRED*]. The whole-genome sequenced strain *vc20357* was outcrossed to *him-5(e1467)* three times and the *C06G1.1* locus was sequenced to confirm that the slim body phenotype is linked to the missense gk306193 mutation (D48N).

## Chemotaxis assays and isolation of *obi* mutants

We X-ray mutagenized wild-type *P. pacificus* PS312 (California) at 325 Roentgen for 25 min with radioactive beryllium and picked 40 $P_o$ onto individual plates. We subsequently picked 1067 $F_2$s from 400 $F_1$s and screened 713 viable $F_3$ populations for insensitivity to 150 nl of oriental beetle pheromone, Z-7-tetradecen-2-one (99% pure; Bedoukian Research, CT) after 1 hr treatment with 1 mM exogenous 8-Bromo-cGMP (Sigma–Aldrich, MO) (*Hong et al., 2008a*). 100 mM nicotinic acid (Sigma–Aldrich, MO) was dissolved in water. Population chemotaxis assays were conducted for ~16 hr at 23°C, and individual candidate F3 from plates showing insensitivity were rescued onto fresh plates by mouth pipetting. We repeated the cGMP-treated chemotaxis assay from the progeny of the candidate mutant lines two more times with at least two duplicate assay plates per line. Only those lines that failed the third round of chemotaxis assay were considered Oriental Beetle pheromone Insensitive *obi* alleles. *Ppa-obi-1* maintained long body morphology and *obi-3* displayed more coiled body posture after 3× outcrossing to wild-type PS312 and selecting the ZTDO insensitive phenotype. Because both the chemotaxis and morphological phenotypes co-segregate after 5× outcross, we subsequently used the long body phenotype as a visible marker for *Ppa-obi-1* during positional mapping. Population chemotaxis assays were performed as previously described based on modified protocol for *C. elegans* (*Bargmann et al., 1993*; *Hong and Sommer, 2006a*). Well-fed J4 and adult worms were washed in M9 buffer twice before assay. For a more consistent response, *C. elegans* was washed in spring water rather than M9 buffer before the ZTDO avoidance assay. For the exogenous cGMP treatment, each washed population was split for mock or 1 hr treatment with 1 mM 8-Bromo-cGMP (in water). For each condition, at least 10 worms had to reach the odor sources in each assay and 12 assays (sample size *N*) were performed over three sessions. Chemotaxis assays with transgenic *obi-1* rescue worms were scored by *rfp* expression in the tail.

## ZTDO paralysis assays

20 live worms or eggs from non-starved cultures were placed on NGM agar plates with fresh OP50. For consistency, eggs were synchronized by picking ~50 adults onto OP50 and allowing them to lay eggs for 3 hr. The eggs were then transferred to different plates and counted. 10 µl of ZTDO or ETDA diluted in ethanol was added to the lid of the plate, immediately sealed with Parafilm, and incubated at room temperature. Larvae were scored 3 hr later or the next day by counting the number of paralyzed and active worms. A worm was scored as paralyzed if it did not move at least one pharynx length after tapping on the outside of the plate. Eggs were scored 2 days later by counting the number of unhatched eggs. For dauer larvae exit assays on NGM plates containing none, 0.001% or 0.01% ZTDO (vol/vol), 25 dauer larvae from the same 2- to 3-week old starved plate were picked onto each type of plate with OP50 and scored for arrested/active dauer larvae, J3 larvae, or J4 larvae after 2 days at ~23°C.

## Whole genome analysis

A single library was paired-end sequenced twice in separate flow cell lanes with 2 × 76 bp that yielded 61 million and 65 million reads after preprocessing (quality control, adapter removal). More than 90% of all reads could be mapped to the *P. pacificus* genome (www.pristionchus.org) with BWA (version 0.5.9) for each library. We then employed ACCUSA for SNP detection (*Froehler and Dieterich, 2010*). Briefly, we performed a head-to-head comparison of the mutant *Ppa-obi-1* and wild-type PS312 mapped libraries. The algorithm represents read stacks (base calls and qualities thereof) as Dirichlet distributions. For any given genome position, we require a read coverage of at least three in both samples. The smaller read stack serves as reference distribution from which the larger stack is drawn. A probability density for this draw is computed, and a SNP is called if the log-density is lower than −30.

## Transgenic reporter gene constructs

Stable transgene expression in *P. pacificus* requires co-injection of endogenous genomic DNA and co-injection markers containing compatible cohesive ends with the target gene as previously described

(*Schlager et al., 2009*; *Cinkornpumin and Hong, 2011*). A 4.3 kb *Ppa-obi-1* promoter fragment was amplified from PS312 gDNA using RHL121/RHL136 (primary) and RHL124/RHL149 (secondary) primers. The *Ppa-obi-1* promoter was then fused with the gfp- *Ppa-rpl-23-3'UTR* by PCR (*Hobert, 2002*). PS312 animals were injected with a DNA mix containing a final concentration of 10 ng/µl of the SpeI/SalI digested 4.3 kb *Ppa-obi-1 promoter::NLS-gfp*-SalI fusion PCR product from three independent reactions; 10 ng/µl SpeI digested *Ppa-egl-20p::rfp* linearized plasmid; and 60 ng/µl SpeI/SalI digested PS312 genomic DNA. Individual $F_1$s expressing *Ppa-egl-20p::rfp* in the hermaphrodite tail were selected for *Ppa-obi-1* expression analysis. Two stable transgenic lines were initially generated with similar expression patterns. We used the stably transmitted reporter line csuIs01[*Ppa-obi-1p::gfp; Ppa-egl-20p::rfp; PS312 gDNA*]X for detailed expression analyses. *csuIs01* is likely to be the result of a spontaneous integration into the X chromosome based on the observation that transgenic males expressing the co-injection *rfp* marker crossed to *Ppa-unc-1*(ChrIV) hermaphrodites gave rise to all $F_1$ males lacking *rfp* expression but all $F_1$ hermaphrodites expressing *rfp*. More than 50 individuals at all developmental stages were analyzed. Ventral views of the vulva expression were obtained from *csuIs01* in the *prl-1* roller background. The same 4.3 kb *Ppa-obi-1* promoter was used to drive the full length 1.6 kb cDNA fused to GFP by injecting into PS312 and crossing into the *Ppa-obi-1(tu404)* background to generate *csuEx27[Ppa-obi-1p::Ppa-obi-1 cDNA:gfp; Ppa-egl-20::rfp; Ppa-obi-1(tu404) gDNA]*. For unknown reasons, the N-terminally fused *gfp* coding region did not express GFP.

To create the *C06G1.1p::gfp* reporter gene construct, we PCR amplified a 3.2 kb upstream sequence from the C06G1.1 ATG using RHL169 and RHL187 and fused the promoter with GFP:*Ppa-rpl-23* 3' UTR by fusion PCR, using RHL181 and RHL154 primers to obtain a 4183 bp amplicon. Independently amplified *C06G1.1p::gfp* fragments were co-injected with either *pRF4 rol-6(su1006)* or *myo-2p::DsRED* vectors along with the pBluescript vector as carrier DNA (50 ng/µl each). This reporter likely encompasses *C06G1.1*'s maximum regulatory region because the 3.2 kb promoter fragment extends ~300 bp upstream of the next gene on the opposite strand (*C06G1.14*), and *C06G1.1* does not contain any intron larger than 1 kb that may act as potential intronic enhancers. Selection for DsRED animals in the reporter strain *csuEx02* [*C06G1.1p::gfp; Cel-myo-2::DsRED*] results in 90% GFP+/DsRED+ progeny and equal DsRED+ expression in pharyngeal muscle cells, suggesting stable mitotic transmission of the transgene. All five F1 transgenic lines showed similar expression profile and intensity in post-hatching stages, but strong body epidermal expression was detected only in the *csuEx03* [*C06G1.1p::gfp; pRF4; pBST*] line. Images were obtained using a Leica DM6000 fluorescent microscope and Leica SP5 confocal microscope. To create the *T02B11.3p::Ppa-obi-1:gfp* amphid sheath expressing translational fusion construct, an ~2.1 kb *T02B11.3* promoter fragment was amplified from *C. elegans* gDNA using RHL476/RHL490 (primary reaction). The *Cel-T02B11.3* promoter is then fused to the same 1.6 kb *Ppa-obi-1* cDNA rescue construct by PCR using RHL477/RHL483 (secondary) to generate a 3.7 kb fusion. This construct and the *GFP-unc-54-3'UTR* fragment, amplified from the pPD95.75 vector using RHL480/RHL481 (primary) was fused using RHL501/RHL482 (tertiary) to create the complete construct. A similar scheme was used to generate *F16F9.3p::Ppa-obi-1:gfp* using RHL498/RHL500 (promoter). *C. elegans* N2 were injected with a mixture of purified *T02B11.3p::Ppa-obi-1:gfp* (25 ng/µl) or *F16F9.3p::Ppa-obi-1:gfp*, *myo-2::DsRED* (25 ng/µl) and pBluescript carrier DNA (100 ng/µl). One stable line from each construct was characterized for ZTDO chemotaxis and paralysis.

The GenBank accession for *Ppa-obi-1* mRNA is JQ946290. Primer sequences are listed in *Table 2*.

## Egg count and body length measurements

Body measurements were obtained as previously described (*Hong et al., 2008a*). In brief, we used a Leica DM6000 and LAS application to measure body dimensions at 24-hr intervals starting at mid-J4/L4 stage. To determine hatching rate, we picked single J4-stage PS312 hermaphrodites onto fresh OP50 plates and transferred the nematodes every 24 hr three times at 20°C. The total number of eggs present on the plates following each transfer and the resulting number of 3-day old animals for each worm during the 3-day period were used to calculate the percentage of hatching and brood size.

## RNA interference

The bacterial feeding strain containing the *C06G1.1* fragment 1 was previously described in a large-scale *C. elegans* RNAi screen (*Simmer et al., 2003*). *C06G1.1* fragment 2 was made using RHL267 and

**Table 2.** Primers

| Primer | Primer sequence (5′→3′) | Template target |
|---|---|---|
| RHL101 | TGAACGGGCTTTTAATCTGG | *Ppa-obi-1* gDNA |
| RHL102 | GTGTCGAGATAGTCGGGCAT | *Ppa-obi-1* gDNA |
| RHL104 | ATGACTGCTCCAAAGAAG | *Ppa-obi-1* cDNA |
| RHL107 | ATTTGCCCTTCGTTCCAC | *Ppa-obi-1* for RNAi |
| RHL108 | CTGTCTCAGTATGCCGAATG | *Ppa-obi-1* for RNAi |
| RHL109 | ATTCAGCCGTGTTACAAATC | *Ppa-obi-1* cDNA |
| RHL110 | CTGCATGTGTTCCGGTCTCG | *Ppa-obi-1* cDNA |
| RHL113 | AAACCTCTGACACATGCAGC | *Ppa-obi-1* cDNA |
| RHL121 | AGTTTCAAAGTTGCATGG | *Ppa-obi-1* promoter |
| RHL124 | CTTCTTTGGAGCAGTCAT**GTAGCGATAATCAGGAGT** | *gfp; Ppa-obi-1* promoter |
| RHL132 | ATGACTGCTCCAAAGAAG | *gfp* |
| RHL136 | CAACCGTGTGCAGTAGAACC | *Ppa-obi-1* 5′ RACE |
| RHL149 | ACTATCCTATCATCGGAAGC | *Ppa-obi-1* promoter |
| RHL151 | AGGGCATTGACAAATCATCG | *Ppa-obi-1* cDNA |
| RHL154 | CACGACGTTGTAAAACGACG | *Ppa-obi-1* cDNA |
| RHL169 | CGCGCTAAACCAATTCCGCC | *C06G1.1* promoter |
| RHL181 | GTTGAGAGAAGAGGTGGAGTCC | *C06G1.1* promoter |
| RHL187 | CTTCTTTGGAGCAGTCAT**CAGATCTGAAAATTTGACACATTG** | *gfp; C06G1.1* promoter |
| RHL267 | GGAACTAACATTAATGTCAC | *C06G1.1* RNAi |
| RHL268 | GAGAAGTGCATAGTTGATG | *C06G1.1* RNAi |
| RHL477 | GCGCTACAGGGGATTTTGTC | *T02B11.3* promoter |
| RHL480 | TGAAAGATGCAAGTAAAGGAGAAGAACTTTTC | p95.75 GFP |
| RHL481 | AAGGGCCCGTACGGCCGACTA | p95.75 GFP |
| RHL482 | GTAGGAAACAGTTATGTTTGG | p95.75 GFP |
| RHL484 | CTTCAGAAAATGATCAGGAGTCTGGTTC | *obi-1* cDNA |
| RHL487 | AGTGAAAAGTTCTTCTCCTTTACT**TGCATCTTTCACGGAGTC** | *obi-1* cDNA; *gfp* |
| RHL490 | AACGAGAACCAGACTCCTGATCAT**TTTCTGAAGAAAGTTGAAAAAC** | *T02B11.3* promoter; *obi-1* cDNA |
| RHL498 | GCGATAAGATCGGTCAATCTGAG | *F16F9.3* promoter |
| RHL499 | GCCAGTAAGGGCTAGTAAGTG | *F16F9.3* promoter |
| RHL500 | AACGAGAACCAGACTCCTGATCAT**ATTTTGTTTCTTACTGTCTTG** | *F16F9.3* promoter |
| RHL501 | ATGGGTACGTTTCTGGGTATAG | *T02B11.3* promoter; *obi-1* cDNA |
| VR86 | GACTCCTGATTATCGCTACG | *Ppa-obi-1* cDNA |
| VR87 | AGATAGTCAAACTATGCATC | *Ppa-obi-1* cDNA |
| VR88 | CTAATAATCTACACTAAATG | *Ppa-obi-1* cDNA |
| AG11112 | CTCGGAGGAGGAACTGGATC | *Ppa-beta-tubulin* RT-qPCR |
| AG11113 | GACCGTGTCAGAGACCTTAG | *Ppa-beta-tubulin* RT-qPCR |
| RH12548 | GCAGGAACATATATTCGCAG | *Ppa-egl-4* RT-qPCR |
| RH12549 | TGTCACGGAACGTTTTGTAC | *Ppa-egl-4* RT-qPCR |
| SL1 | GGTTTAATTACCCAAGTTTGAG | SL1 for 5′ RACE |
| SSLP21F | ACTGGTGCTCATCGCAAGA | Mapping marker |

*Table 2. Continued on next page*

*Table 2. Continued*

| Primer | Primer sequence (5′→3′) | Template target |
|--------|-------------------------|-----------------|
| SSLP21R | CCTTCTGTTTTCCTACCCCC | Mapping marker |
| SSLP7F | TCTTGCATAACACCGAACAAA | Mapping marker |
| SSLP7R | AGGGCGTTACGTGAATAAGC | Mapping marker |
| SSLP17F | CAATGCAGTAGCCGAATCAT | Mapping marker |
| SSLP17R | CAATATTGGCCTTCACCCTG | Mapping marker |
| S447F | TGGGGATAGCGAAAATCATC | Mapping marker |
| S447R | CGCATCGTTATTCACGAAATTC | Mapping marker |

Letters in bold are based on different templates for fusion PCR.

RHL268 primers to amplify an ~400 bp fragment of *C06G1.1*, digesting this product with BglII and PstI, and ligating the 190 bp fragment into the BglII/PstI sites of pL4440. In vitro synthesis of dsRNA from 1 µg of T7 amplified PCR template from the RNAi vector or the pL4440 negative control was accomplished using the Ambion in vitro synthesis kit (Grand Island, NY). RNAi by bacterial feeding of *C06G1.1* was performed by scoring the progeny of L4 stage *rrf-3(pk1426)* animals raised on RNAi or L4440 control plates containing 1 mM IPTG and 50 µg/ml carbenicillin.

## Statistical analysis

Means, SEM (error bars), and p values for two-tailed *t* test were performed by Microsoft Excel. One-way ANOVA with multiple comparisons post-hoc testing were performed using Prism statistical software. Sample sizes represent total technical replicates from two or more biological replicates.

## Acknowledgements

RLH is supported by the NIH grant SC2GM089602, 1SC3GM105579, and the CSUN Research and Scholarship award. JLG was supported by the Keck Foundation. The Caenorhabditis Genetics Center (CGC) provided the *C. elegans* strains. We thank J Salandanan, E Bernal, J Escobedo, and G Aguilar-Portillo for technical assistance, the Shaham lab for plasmids, and J Srinivasan and P Sternberg for help with genome sequencing.

## Additional information

### Funding

| Funder | Grant reference number | Author |
|--------|------------------------|--------|
| National Institute of General Medical Sciences | SC2GM089602 | Ray L Hong |
| Center for California Studies, California State University | Research and Scholarship Award | Ray L Hong |
| W.M. Keck Foundation | | James L Go |
| National Institutes of Health | 1SC3GM105579 | Ray L Hong |

The funders had no role in study design, data collection and interpretation, or the decision to submit the work for publication.

### Author contributions

JKC, DRW, JLG, RLH, Conception and design, Acquisition of data, Analysis and interpretation of data, Drafting or revising the article; VR, CD, Acquisition of data, Analysis and interpretation of data; XW, Acquisition of data, Analysis and interpretation of data, Contributed unpublished essential data or reagents; RJS, Drafting or revising the article, Contributed unpublished essential data or reagents

## Additional files

### Major dataset

The following previously published datasets were used:

| Author(s) | Year | Dataset title | Dataset ID and/or URL | Database, license, and accessibility information |
|---|---|---|---|---|
| Hong RL and Rapoport V | 2013 | A Lipid-binding Protein in the Necromenic Nematode Pristionchus pacificus is Involved in Insect Pheromone Sensing | JQ946290; http://www.ncbi.nlm.nih.gov/nuccore/JQ946290 | Publicly available at GenBank (http://www.ncbi.nlm.nih.gov/genbank). |
| Yook K, et al., | 2012 | WormBase 2012: more genomes, more data, new website | C06G1.1; NRF-5, Caenorhabditis elegan; JA26805, Caenorhabditis japonica; CBP03891, Caenorhabditis briggsae; RP35181, Brugia malayi; doi: 10.1093/nar/gkr954 | Publicly available at WormBase (http://www.wormbase.org/). |
| Ghedin E, Wang S, Spiro D, Caler E, Zhao Q, Crabtree J, Allen JE, Delcher AL, Guiliano DB, Miranda-Saavedra D, Angiuoli SV, Creasy T, Amedeo P, Haas B, El-Sayed NM, Wortman JR, Feldblyum T, Tallon L, Schatz M, Shumway M, Koo H, Salzberg SL, Schobel S, Pertea M, Pop M, White O, Barton GJ, Carlow CK, Crawford MJ, Daub J, Dimmic MW, Estes CF, Foster JM, Ganatra M, Gregory WF, Johnson NM, Jin J, Komuniecki R, Korf I, Kumar S, Laney S, Li BW, Li W, Lindblom TH, Lustigman S, Ma D, Maina CV, Martin DM, McCarter JP, McReynolds L, Mitreva M, Nutman TB, Parkinson J, Peregrin-Alvarez JM, Poole C, Ren Q, Saunders L, Sluder AE, Smith K, Stanke M, Unnasch TR, Ware J, Wei AD, Weil G, Williams DJ, Zhang Y, Willia | 2008 | Draft genome of the filarial nematode parasite Brugia malayi | XP_001897990.1; http://www.ncbi.nlm.nih.gov/protein/XP_001897990.1 | Publicly available at GenBank (http://www.ncbi.nlm.nih.gov/genbank). |
| Singh VV, Chauhan SK, Rai R, Kumar A, Singh SM and Rai G | 2014 | Decreased pattern recognition receptor signaling, interferon-signature, and bactericidal/permeability-increasing protein gene expression in cord blood of term low birth weight human newborns | NP_001716.2; http://www.ncbi.nlm.nih.gov/protein/NP_001716.2 | Publicly available at GenBank (http://www.ncbi.nlm.nih.gov/genbank). |

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
