## [Decision Letter]

Thank you for sending your work entitled “A Host Beetle Pheromone Regulates Development and Behavior in the Nematode *Pristionchus pacificus*” for consideration at *eLife*. Your article has been favorably evaluated by Ian Baldwin (Senior editor), Oliver Hobert (Reviewing editor), and 3 reviewers.

The Reviewing editor and the other reviewers discussed their comments before we reached this decision, and the Reviewing editor has summarized the substantive comments that need to be addressed:

1) Please strengthen the analysis of the role of the amphid sheath cell. Either ablate amphid sheath in Ppa or express Ppa *obi-1* (or C06G1.1) in *C. elegans* under an amphid sheath cell promoter to see if *C. elegans* now becomes sensitive to ZTDO.

2) RNAi of C06G1.1 to resolve its role in *C. elegans*.

3) Please include all data that support the claim that *tu404* and *tu405* are allelic.

4) Strengthen or remove the feeding rescue experiment, which appears weak.

5) Strengthen or more explicitly discuss the caveat of the difference in gene expression of *Cel-obi-1*.

Reviewer #1:

The authors make the interesting observation that the beetle sex pheromone ZTDO a known strong attractant to Pristionchus adults arrests Pritionchus development. Thus ZTDO may thus limit Pristioncus growth upon infection. In a genetic screen they identify a novel gene obi-1 which provides a molecular mechanism for the attraction and sensitivity to the beetle sex pheromone. The role of ZTDO as a developmental regulator are unexpected and exciting and shed new light on the evolution of nematode-insect associations. The data are clearly presented.

I only have a few suggestions that could improve the manuscript and substantiate their model:

1) The authors hypothesize that the differential expression between C.e C06G1.1 and Ppa-obi-1 may explain the difference between the susceptibility to ZDTO

They dismiss the possibility that changes in protein function unlikely to explain the difference between obi-1 and C06G1.1 (line 300)? To me 61% shared identity suggests there is plenty of protein difference. The authors favor a difference in expression pattern to explain the difference between *C. elegans* and P.pa response to ZDTO. A third possibility is that there is another specific receptor for ZDTO that interacts with obi-1 in Ppa but not in *C. elegans* that the authors could mention. The authors focus on the obi-1 expression in glia like sheath cells in Ppa but not *C. elegans*. They suggest that glia like sheath cells expression of obi-1 in Ppa could explain the difference in chemosensation. To substantiate their hypothesis a feasible experiment would be to express obi-1 under control of the C06G1.1 promoter in *C. elegans* and test for attraction and paralysis.

2) It would have been interesting to determine the null phenotype of the C06G1.1 in *C. elegans*. The authors did analyze a missense mutation for C06G1.1. I don't think it is clear what the effect of the D48N C06G1.1 missense mutation is. More useful might be a C06G1.1 RNAi experiment. In wormbase the RNAi phenotype is long like the obi-1 mutant in Pristionchus suggesting that RNAi works for C06G1.1. Is D48N C06G1.1 long? According to the authors one would expect that C06G1.1 RNAi animals are more sensitive to the paralytic effect of ZDTO. This should be a straight forward and easy experiment.

If DL is the primary infective stage in many entomophilic nematodes found on live beetles one would expect ZTDO to be attractive to DL. Have the authors determined the chemotaxis response of DL at even lower concentrations ZTDO?

Not suggesting that all experiments should be done but some of these experiments might substantiate their conclusions.

1) Can the authors specify what percentage of arrested Dauer Larva are paralyzed rather than some and majority? Is the developmental block specific for dauer exit or do they not exit the dauer stage because of paralysis?

2) I did not completely follow the rationale for the *egl-4* experiment is not entirely clear. Is the role of *egl-4* in chemosensation and body size important? Also reference Etoille 2002 should that be Daniels 2000?

3) No egg laying data are presented so show the data or add “(data not shown)”

4) How do you know they are in equal abundance? RT-PCR? qPCR? “(data not shown)”.

5) Since splice form variation does not contribute to the difference in insect pheromone attraction, not sure why this is “Interestingly”.

6) Why are the data in Figure 4 different from those in Figure 1?

Reviewer #2:

Cincornpumin and colleague report the discovery of a molecule required for the detection of a host-derived cue and its effects on the development of the necromenic nematode Pristionchus pacificus. Pristionchus is a powerful model for comparative molecular studies of development and behavior, and also offers the opportunity to study the molecular bases of interspecies interactions because of its fascinating ethology. This work is rigorous and clearly presented. The findings not only illuminate an important molecular mechanisms that functions in Pristionchus but also has important implications for the community of researchers who use C. elegans as a model to study chemosensation. I have one issue that I feel should definitely be addressed by the authors before publication and several suggestions that might strengthen the story.

Two alleles of *obi-1* – *tu404* and *tu405* – were isolated by screens for mutants defective in beetle pheromone sensing. The authors refer to data from complementation testing but do not show these data. Please include all data that support the claim the *tu404* and *tu405* are allelic. Also, the authors only report the molecular characterization of tu404. Is there no coding mutation in the *tu405* strain? Does the transgene that rescues the *tu404* mutant phenotype also rescue the *tu405* mutant phenotype? In its current form the data in the manuscript leave open the possibility that *tu404* and *tu405* are not allelic but display some interesting interactions.

Reviewer #3:

I like the results described in this manuscript. The finding that beetle odor inhibits growth of Pristionchus at three stages: embryo development, paralysis of J2 and blocking dauer exit is very interesting. The screening for and positional cloning of the *obi-1* gene represents a significant and important effort. The cloning is quite convincing, with transgenic rescue following fine genetic mapping and sequencing. OBI-1 is a putative lipid binding protein expressed in potential secretory cells. I have two major concerns;

1) One important point made by the authors is the difference in expression between Ppa and Cel. However, the expression pattern in *C. elegans* is based solely on a transgenic reporter assay. Since this is a key point, I think smFISH of Cel-C06G1.1 (*Cel-obi-1*) is necessary to establish a convincing negative expression result.

2) I am confused and concerned by the feeding rescue experiment. It just seems weird to me. I suppose it could be acting in the intestine. This result needs shoring up with something like localized expression of *obi-1* in gut, nervous system, etc.

---

## [Author Response]

*1) Please strengthen the analysis of the role of the amphid sheath cell. Either ablate amphid sheath in Ppa or express Ppa obi-1 (or C06G1.1) in C. elegans under an amphid sheath cell promoter to see if C. elegans now becomes sensitive to ZTDO*.

We constructed *C. elegans* transgenic strains that expressed *Ppa-obi-1* cDNA under the amphid sheath cell-specific promoter *T02B11.3* and *F16F9.3*. The translational fusion protein *Ppa*-OBI-1:GFP is expressed faithfully in amphid sheath cells and improved the ability of *F16F9.3p::obi-1:gfp* worms to resist ZTDO-induced paralysis in the L4 larvae (Figure 5—figure supplement 2). However, misexpression of *Ppa*-OBI-1 in the *C. elegans* amphid sheath cells did not result in changes in chemotaxis response to ZTDO. We attempted but were not able to train a researcher to use a cell ablation setup in a facility nearby to definitively answer if amphid cells mediate ZTDO sensitivity in *P. pacificus*. We have thus tempered our speculation that *Ppa*-OBI-1is required only in the amphid sheath cells in *P. pacificus*. However, given the growing evidence by the Shaham lab at the Rockefeller University that glial cells are important for amphid neuron ontogeny and function in *C. elegans*, and that secreted proteins in the sheath cell have not been actively studied, we feel obliged to discuss several models for how *Ppa*-OBI-1 proteins may function in the amphid sheath cells.

*2) RNAi of C06G1.1 to resolve its role in C.* elegans.

We have performed RNAi against *C06G1.1* in *C. elegans* using two fragments of the coding sequence and measured their body dimensions (Figure 4—figure supplement 4). RNAi resulted in longer and slenderer body, consistent with the reported RNAi phenotype in www.wormbase.org based on results published by [43].

*3) Please include all data that support the claim that tu404 and tu405 are allelic*.

We have made a typographical error and meant to indicate that *Ppa-obi-1(tu404)* and *Ppa-obi-3(tu405)* are *not* allelic because they *do* complement each other in the F1 cross. This passage was corrected in the main text.

*4) Strengthen or remove the feeding rescue experiment, which appears weak*.

We have removed the feeding rescue experiment due to the weak effects.

*5) Strengthen or more explicitly discuss the caveat of the difference in gene expression of Cel-obi-1*.

We acknowledge that small changes in amino acid sequence can have profound effects on protein function, so we have removed the statement that suggests changes in gene expression pattern is more likely than changes in protein function to contribute to the differences in ZTDO sensitivity between *C. elegans* and *P. pacificus*. The statement now reads “While small changes in sequence can lead to large changes in protein function, the absence of expression of the *C. elegans obi-1* ortholog in amphid sheath cells suggests changes in gene expression pattern may also play a role in determining ZTDO sensitivity.” We also mentioned other possible tissues types that can mediate ZTDO sensing.

Reviewer #1:

1) Can the authors specify what percentage of arrested Dauer Larva are paralyzed rather than some and majority? Is the developmental block specific for dauer exit or do they not exit the dauer stage because of paralysis?

Very few wild-type DL became paralyzed or died on plates with 0-0.001% ZTDO. Approximately 40% wild-type DL remained active after two days on food with 0.01% ZTDO (Figure 2). However for the *Ppa-obi-1* mutant, the dauer-exit defect is exacerbated by the hypersensitivity to ZTDO, in which 79±17% of all two-day old dauers became paralyzed and died on plates without ZTDO, and 94.6±5% DL became paralyzed on 0.001% ZTDO. Therefore, 0.01% ZTDO can block active wild-type *P. pacificus* DL from progressing to the reproductive stage without causing paralysis, but 0.001-0.01% ZTDO significantly reduced dauer exit in *Ppa-obi-1* DL primary by inducing paralysis and death.

We have explained this finding more explicitly in the main text.

*2) I did not completely follow the rationale for the egl-4 experiment is not entirely clear*. *Is the role of egl-4 in chemosensation and body size important? Also reference Etoille 2002 should that be Daniels 2000?*

We have added the reference to Daniels 2000 regarding *egl-4*’s role in dauer development. Daniels 2000 showed that *C. elegans egl-4* loss-of-function allele *n479* is daf-c (dauer constitutive), whereas we detected a weak but statistically significant dauer exit phenotype in the *P. pacificus egl-4(tu374)* allele (Figure 2).

3) No egg laying data are presented so show the data or add “(data not shown)”

We have added the egg-laying data for *Ppa-obi-1(tu404)* as Table 1.

4) How do you know they are in equal abundance? RT-PCR? qPCR? “(data not shown)”.

We subcloned *Ppa-obi-1* cDNA from multiple RT-PCR products using primers from the first and last exons and sequenced at least 6 clones from both strains. We found 3 inserts of each short and long splice variants.

5) Since splice form variation does not contribute to the difference in insect pheromone attraction, not sure why this is “Interestingly”.

We have reworded this passage to “In the two *Ppa-obi-1* splice forms, the longer transcript contains a VKDDDPR peptide that may be the result of an alternative splice donor site. Both splice variants were isolated from the California and Washington strains in equal abundance by subcloning RT-PCR products (data not shown) and no differences in nucleotide sequence were found in the coding region of these two strains.”

6) Why are the data in Figure 4 different from those in Figure 1?

The results for *Ppa-obi-1(tu404)’s* neutral response to ZTDO in Figure 1 were obtained in Germany 6 years ago, while *Ppa-obi-1’s* avoidance response in Figure 4 was obtained in the PI’s new lab at California State University Northridge. This more recent result is highly repeatable and was also confirmed by testing animals from a frozen *Ppa-obi-1* strain. The ZTDO chemical supplier remained the same. Thus, it is possible that unlike wild-type PS312 (whose response to ZTDO remained unchanged), *Ppa-obi-1* is also responding to differences in other chemical compositions (e.g. water or media) in the current lab environment. All chemoattraction assays were performed with concurrent controls and indicated successful transgenic rescue of *Ppa-obi-1(tu404)*.

Reviewer #2:

*Two alleles of obi-1 – tu404 and tu405 – were isolated by screens for mutants defective in beetle pheromone sensing. The authors refer to data from complementation testing but do not show these data. Please include all data that support the claim the tu404 and tu405 are allelic. Also, the authors only report the molecular characterization of tu404. Is there no coding mutation in the tu405 strain? Does the transgene that rescues the tu404 mutant phenotype also rescue the tu405 mutant phenotype? In its current form the data in the manuscript leave open the possibility that tu404 and tu405 are not allelic but display some interesting interactions*.

The typographical error was corrected as mentioned previously. *Ppa-obi-1(tu404)* and *Ppa-obi-3(tu405)* are *not* allelic because they *do* complement each other in ZTDO attraction. *Ppa-obi-3(tu405)* also does not share the body proportion and ZTDO hypersensitivity phenotypes of *Ppa-obi-1(tu404).*

Reviewer #3:

*1) One important point made by the authors is the difference in expression between Ppa and Cel. However, the expression pattern in C. elegans is based solely on a transgenic reporter assay. Since this is a key point, I think smFISH of Cel-C06G1.1 (Cel-obi-1) is necessary to establish a convincing negative expression result*.

We agree that transcriptional reporters based on a single promoter fragment may not comprehensively represent endogenous mRNA expression patterns. Due to the technical expertise needed to perform the suggested smFISH experiments for *C06G1.1*, we hope to address this caveat in our future research.

*2) I am confused and concerned by the feeding rescue experiment. It just seems weird to me. I suppose it could be acting in the intestine. This result needs shoring up with something like localized expression of obi-1 in gut, nervous system, etc*.

We have removed the feeding rescue experiment.